# Rethinking Local Economic Development for Fetal Alcohol Spectrum Disorder in Renosterberg Local Municipality, South Africa

**DOI:** 10.3390/ijerph20054492

**Published:** 2023-03-03

**Authors:** Bianca Jordan, Naiefa Rashied, Marius Venter

**Affiliations:** 1School of Economics, University of Johannesburg, Johannesburg 2006, South Africa; 2Centre for Local Economic Development, University of Johannesburg, Johannesburg 2006, South Africa

**Keywords:** local economic development, Renosterberg Local Municipality, fetal alcohol spectrum disorder, drinking motives, alcohol consumption

## Abstract

Two towns in Renosterberg Local Municipality (RLM) in the Northern Cape Province of South Africa, Petrusville and Philipstown, have high Fetal Alcohol Spectrum Disorder (FASD) prevalence rates. FASD is linked to poverty and imposes high national economic costs. Thus, it is critical to understand the local economic development (LED) strategies used to mitigate the high prevalence of FASD. Moreover, there is sparse literature examining adult communities where FASD children reside. Understanding these adult communities is important because FASD cannot exist without adult gestational exposure to alcohol consumption. Using a mixed-method approach, this study uses a six-phase analytic approach to investigate the drinking culture and drinking motives in RLM, applied to two cross-sectional in-depth community needs assessments, five in-depth interviews, and three focus groups. This study also evaluates how the RLM targets FASD, as well as binge and risky drinking, in its municipal economic strategy by analysing its Integrated Development Plan (IDP) with respect to an eight-stage policy development process. The results indicate that 57% of respondents expressed concern regarding the unhealthy drinking culture in RLM, 40% felt that the residents of RLM drank in response to unemployment-related hopelessness, and 52% attributed the drinking culture to a lack of hobbies and recreational opportunities. The results of an analysis of the RLM IDP through the lens of Ryder’s eight-stage policy development process suggest that the decisive policy development process is not open to the public and that, furthermore, FASD is neglected. A dedicated alcohol consumption census-style study is recommended to broadly capture alcohol consumption in RLM, allowing researchers to identify the exact alcohol consumption patterns and priority areas for the IDP and public health policy. RLM should directly publicise its policy development process so that its IDP is inclusively formulated to address FASD, risky drinking, binge drinking, and gestational alcohol consumption.

## 1. Introduction

Alcohol is an important part of various cultural and recreational practices. However, it has also been linked to several harmful behaviours [1]. The use of alcohol during pregnancy is one such harmful behaviour. Globally, the estimated prevalence of alcohol use during pregnancy is 9.8% [1]. Alcohol consumption during pregnancy can create several challenges for infants, such as Fetal Alcohol Spectrum Disorder (FASD) [2]. FASD affects many Coloured, otherwise known as people of mixed heritage, communities across South Africa’s Cape region, a remnant of the dop system of apartheid-era South Africa [3]. The dop system (dop is the Afrikaans word for an alcoholic drink) is a system where alcoholic beverages were offered to farmworkers as their weekly wage instead of payment [4]. South Africa has reported FASD prevalence rates ranging from 6.7 per 1000 Grade 1 children in the Saldana Bay Local Municipality (SBLM) to 282 per 1000 Grade 1 children in the Renosterberg Local Municipality (RLM) [5,6,7]. The literature on FASD in South Africa focuses predominantly on childhood prevalence. There is sparse literature examining adult communities where FASD children reside. Understanding these adult communities is important because FASD cannot exist without adult gestational exposure to alcohol consumption.

Two towns in RLM in the Northern Cape Province of South Africa, namely Petrusville and Philipstown, have alarmingly high reported prevalence rates of FASD [3]. Moreover, the socioeconomic conditions in RLM are bleak. Most males seek employment outside the RLM in other municipalities or provinces in South Africa [8]. Women have limited economic opportunities for sustainable livelihoods [8]. From a local government perspective, RLM is classified as one of the ten least efficient municipalities in South Africa [9].

Local Economic Development (LED) is defined as an adaptive and responsive process where various partners work collectively to improve economic conditions for innovation-driven inclusive economic development in a specific area, in this case at the municipal level. Inclusive economic development is characterised by the achievement of knowledge transfer, employment generation, competence building, and revenue generation in a local area in order to improve its economic future and the quality of life for its inhabitants [10]. LED offers solutions to address socioeconomic challenges in local municipalities. There are many examples of successful applications of the LED approach, one of which is the case of a waste management initiative in Mutare, Zimbabwe [11]. The National Framework for Local Economic Development 2018/2028 (NFLED) in South Africa was developed to stimulate and cultivate a deeper understanding of LED [10]. The NFLED aims to provide strategic guidance to the key players and decision makers to identify local opportunities to adequately address local needs.

The linkage between poverty and mothers of children with FASD has been documented in the literature [12]. For example, women with lower socioeconomic status have a higher risk of bearing children with FASD [13,14]. Moreover, women with FASD children have lower levels of education and income and are employed far less frequently [13,14]. There are linkages between higher economic status and mothers of children with FASD [15]. However, these studies were conducted in developed countries such as Spain [16], the United Kingdom [17], and Australia [18]. FASD imposes high economic costs for the government and individuals [19]. For example, in Canada, FASD imposes both direct costs, e.g., screening, diagnosis, childcare, and special education, and indirect costs, e.g., productivity losses due to premature mortality, morbidity, and law enforcement, on the healthcare system. In 2013, the total overall FASD-related costs in Canada ranged from CAD 1,287,707,594 to CAD 23,412,655,151 [20]. Therefore, it is important to understand the LED strategies that can be used to mitigate the high prevalence of FASD. The aim of this study is to explore the socioeconomic status, drinking culture, and drinking motives of residents in RLM, as well as the current municipal economic strategies, to provide practical recommendations for future consideration.

## 2. Materials and Methods

### 2.1. Study Setting and Dataset

Data sources for the RLM are relatively sparse. As a result, this study makes use of multiple data sources. A mixed-method approach is defined as analysing both quantitative and qualitative primary and/or secondary data [21,22]. The study uses a mixed-method research design through an integrated analysis of various qualitative and quantitative primary and secondary data, as indicated in Table 1, some of which are not publicly available. The use of primary and secondary data sources received ethical clearance from the authors’ university research ethics committee and is outlined further in the Ethics Committee Statement at the end of this paper.

Situated on the border with the Free State, Petrusville is the largest town in the RLM and is one of eight within the Pixley ka Seme district. The second largest town is Philipstown, followed by Vanderkloof [23]. For the primary data sources, applied convenience sampling was used for Petrusville and Philipstown, as this was logistically more manageable and cost effective. As a result, the study excluded the Vanderkloof population of 1228 people [23].

### 2.2. Instruments

The data collection instruments for the primary research component of the study consisted of focus groups and telephonic, structured, in-depth interviews. Due to COVID-19 pandemic restrictions, all primary research was undertaken virtually by the first author using the Microsoft Teams 2021 (Microsoft Teams is a messaging app used in organization workspaces for collaboration and communication in real time, as well as for meetings and file and app sharing) and Zoom 2021 (Zoom is a video conferencing software app that enables face-to-face communication when in-person meetings are not possible) software applications. A guiding questionnaire was used as a measure to ensure relevant and useful responses. Any community member from Petrusville or Philipstown was eligible to participate in the focus groups. Key stakeholders operating in the area were eligible for the in-depth interviews and were approached to participate by two community members. Only consenting community members and stakeholders were eventually included in the focus groups and in-depth interviews. Written informed consent was obtained from all participants involved in the study. The same two community members assisted with obtaining written consent from each participant. No formal training was necessary since the two community members had previous experience and training in primary data collection and the related ethics requirements for participation consent. We used open coding in Microsoft Excel 2021 (Microsoft Excel is a versatile spreadsheet software programme used for data visualisation and analysis) to develop our code for the initial themes and terminology in line with phase two of the six-phase analytic process [26]. We did not predefine the codes but rather determined the codes based on the responses received. Thereafter, we condensed the themes into larger groups based on the terminology provided by the participants. Two of the three authors conducted the open coding separately and compared the coding thereafter to confirm the suitability and reliability of the assigned code.

Secondary data were collected by the original researcher, who we refer to as Company A. Company A is a private company operating in RLM. Since the secondary data are not publicly available, we again applied for the relevant ethical clearance through our institution as outlined in the Ethics Committee Statement at the end of this document. Company A used various instruments to collect the data such as surveys, structured interviews, semi-structured interviews, questionnaires, and focus groups. The key motivation for utilising multiple data sources was to ensure that the interpretation of the results could be triangulated using two or more data sources in order to corroborate our research findings [27].

### 2.3. Univariate and Qualitative Analyses

This study began with a univariate, spatial analysis of RLM using census data related to infrastructure, education, employment, and household dynamics to gain in-depth insights into its socioeconomic conditions. Thereafter, the drinking culture and drinking motives in RLM were investigated by applying the six-phase analytic process to two cross-sectional in-depth community needs assessments, five in-depth interviews, and three focus groups [20]. This study also examined how the RLM targets FASD, as well as binge, risky and gestational drinking, in its municipal economic strategy by thematically analysing its IDP using Ryder’s 8-stage policy development process [28].

## 3. Results

The results of this study are divided into three sections. The first section describes the univariate spatial comparative analysis of RLM and SBLM using data sources 5–6, whereas the second section thematically explores the drinking motives and trends in RLM using data sources 1–4 and 7. The final section examines how the RLM targets FASD, as well as binge and risky drinking, in its municipal economic strategy.

### 3.1. Univariate Spatial Comparative Analysis of RLM and SBLM

The results of the univariate spatial analysis are summarised in Table 2, Table 3, Table 4, Table 5 and Table 6. The purpose of this section is to provide the context of the socioeconomic characteristics of the RLM. Where data were available, we included comparisons to the Saldana Bay Local Municipality (SBLM), which has the lowest reported FASD prevalence rates in South Africa, to demonstrate the differences in the economic characteristics [3].

Table 2 indicates that the number of households in RLM increased by 568 between 2011 and 2016. Although the average household size remained constant, the number of formal dwellings decreased by 9.6%, whereas the number of houses owned increased by 15.8%. The percentage of female-headed households was almost double the national average of 16.9%. Table 2 also shows that the number of households in SBLM increased by 555 between 2011 and 2016. Although the average household size remained consistent at 0.1%, the number of formal dwellings decreased by 6.9%, whereas the number of houses owned increased by 10.3%. The percentage of female-headed households in SBLM also exceeded the national average by 16%. This table shows that RLM and SBLM were very similar with respect to household dynamics, except for the number of households, despite SBLM having a substantially lower reported FASD prevalence rate than RLM.

Table 3 provides an overview of the labour market in RLM and SBLM over a 10-year period. Unemployment increased by 4.17% in RLM and 9.53% nationally between 2011 and 2021. In the 10-year period, the employment rate decreased by 1.6% in RLM and 4% nationally. Even though unemployment increased by 8.18% in SBLM, it was still far below the national unemployment rate of 33.98%.

Table 4 shows that RLM and South Africa experienced positive growth with respect to completed secondary education between 2011 and 2016. Higher education attainment increased nationally by 2.8% but decreased in RLM by 1.8%. No schooling increased in RLM by 4.8%. These statistics do not sum to 100 as they exclude the primary education statistics. Table 4 shows that SBLM had a lower rate of no schooling, with slightly higher rates of completion for secondary and higher education.

Table 5 shows that access to a flushing toilet connected to a sewage system increased in both RLM and nationally. Access to other infrastructural amenities, such as weekly refuse removal, piped water inside the dwelling, and electricity for lighting, unfortunately, decreased in RLM over the reported 10-year period. Table 5 shows that SBLM had a higher rate of access to sewerage, weekly refuse removal, and piped water inside the dwelling.

The majority of the workforce in RLM is employed in the agriculture, hunting, and forestry sectors. Since agriculture is the main economic sector in RLM, it employs most of the labour force. The second-largest employment sector is community, social, and personal services, which consists largely of government employees from the local municipality and other government departments. This indicates that job opportunities are primarily in agriculture.

With respect to education, each town in the RLM has limited early childhood development (ECD) centres and primary and secondary schools. With respect to health, each town has only one clinic, which is inadequate for the population size, as outlined in Table 3. Although the number of sports fields and community halls provide some support for recreational activity, the lack of transport networks makes it difficult to commute and participate in recreational activities out of town. Even though they serve a larger community of people, SBLM contains more schools, healthcare facilities, access to transport, and community infrastructure. In addition, SBLM has opportunities for growth through the IDZ and residents have more options for recreation. Table 2, Table 3, Table 4, Table 5 and Table 7 show that compared to SBLM, fewer opportunities for education, a lack of access to basic infrastructure, higher rates of unemployment, and fewer recreational opportunities in RLM can be associated with its higher FASD prevalence rate, although this would need to be confirmed statistically, which is beyond the scope of this study.

Figure 1 represents the results from data source 2. Company A interviewed a representative sample of 50 people from both Petrusville (*n* = 25) and Philipstown (*n* = 25). The purpose of the interviews was to gather baseline data that would inform a community development strategy. The sample consisted of 24 males and 26 females. The racial composition of the sample was 19 African, 29 Coloured, and 2 White South Africans.

The survey participants were asked to complete a survey that consisted of two questions, namely (a) “name three things which you like about your town”, and (b) “name three things which you do not like about your town”.

Figure 1 summarises the key themes identified from the responses to the first question. The category titled “nothing” indicates that community members felt that there was nothing to like about their respective towns. A total of 23% of community members felt a sense of community and belonging towards their town, whereas 16% and 14%, respectively, appreciated the low crime rate. The business opportunities in RLM were appreciated by 14% of community members.

According to Figure 2, 30% of respondents did not appreciate the high levels of alcohol and substance abuse, 40% did not appreciate the poor education and skill levels in RLM, and 28% did not appreciate the lack of constructive and recreational activities.

### 3.2. Drinking Culture, Drinking Motives, and FASD

#### 3.2.1. Drinking Culture

Based on the quotations presented in Table 8, excess alcohol intake begins at a very young age in RLM. Prenatal and gestational alcohol consumption is reported as the norm, with some believing that gestational alcohol consumption is beneficial for fetal development. People who are economically vulnerable have access to cheaper sources of alcohol and there is ample alcohol available for sale from households if one cannot purchase from a licensed liquor outlet. People drink to cope with their adverse socioeconomic or other circumstances and people encourage each other to drink. We applied a six-phase analytic process to analyse the themes from the community needs assessment, in-depth interviews, and focus groups [20].

#### 3.2.2. Drinking Motives

The results of data sources 1, 3, 4, and 7 demonstrate the drinking culture in RLM. Table 9 provides an overview of the drinking motives in RLM.

There are limited after-school recreational activities for children. As a result, increased levels of boredom perpetuate the high rates of alcohol consumption among underage youth. There are few role models available in the community to inspire youth with respect to their career and life aspirations. Increased alcohol use and limited employment prospects also perpetuate the high rates of alcohol consumption. Alcohol is easily available on credit, which makes it easily accessible to those who cannot afford it. Spouses force each other to drink. The table also highlights respondents’ concerns related to the role of alcohol in gender-based violence, rape, and other crimes.

The thematic responses showed that 57% of respondents expressed concern regarding the unhealthy drinking culture in RLM, 40% felt that residents of RLM drank in response to unemployment-related hopelessness, and 52% attributed the drinking culture to a lack of hobbies and recreational opportunities. These percentages were generated by thematically grouping the responses from the focus groups. The percentages were then divided by the total number of respondents to the survey (*n* = 37).

The drinking culture and drinking motives outlined in Table 10 are linked to the views on FASD in Petrusville and Philipstown. The key results, as outlined in Table 10, illustrate that although training and awareness programs about FASD have been implemented in both Petrusville and Philipstown, this does not prevent mothers from drinking during pregnancy. Community views on FASD (commonly termed FAS by community members) illustrate that an alternative intervention is required in RLM since awareness training has a limited influence on changing drinking patterns during pregnancy.

### 3.3. Ryder’s Policy Development Process

The 2018/19 RLM IDP [23] is the only publicly available municipal document in RLM. Therefore, the analysis of the RLM IDP suggests that the decisive policy development process is not open to the public and that, furthermore, FASD is neglected. Table 11 utilises Ryder’s eight-stage policy development process [28] as a guideline for analysing the 2018/19 RLM IDP.

A significant constraint on the development of the IDP is that most senior management of the municipality are temporarily employed, which causes a significant delay with respect to decision making and the implementation of the IDP in RLM. In addition, the IDP process is still regarded as new to many officials joining the municipality, which means that they have very little understanding of its purpose. Lastly, previous management failed to impart skills to junior officials to understand and implement the IDP [23]. In addition, there was no input to the IDP from the relevant government departments.

The results shown in Table 11 indicate that RLM has followed a clear approach to understanding its challenges. However, due to certain limitations, inputs from key government departments are not reflected in the IDP. Therefore, alcohol consumption and FASD are neglected in the IDP.

## 4. Discussion

The results illustrate that there is a strong interconnectedness of the challenges that RLM is facing. RLM is characterised by adverse socioeconomic conditions, as indicated in the results of the univariate spatial analysis. Compared to SBLM, which has the lowest reported FASD prevalence rate in South Africa, the findings from the univariate analysis show that fewer opportunities for education, a lack of access to basic infrastructure, higher rates of unemployment, and fewer recreational opportunities in RLM can be associated with its higher FASD prevalence rate, although this would need to be confirmed statistically, which is beyond the scope of this study.

Our findings are comparable with the existing literature. A previous study illustrated that female-headed households are predominately supported by grants and part-time employment [8]. Female-headed households in RLM are constantly experiencing shortages of money, food, clothing, and school supplies [8]. In addition, most female heads of households in RLM are unemployed and often rely on informal jobs to support their families. The association between poor socioeconomic conditions and mothers of children with FASD has also been documented in the literature. Women with lower socioeconomic status have a higher risk of bearing children with FASD [13,14]. Moreover, women with FASD children have lower levels of education and income and are employed far less frequently.

The results of the focus groups illustrate that the community in RLM understands that alcohol consumption adversely affects families and households. Besides the health risks, excessive alcohol use also contributes to other social problems such as high alcohol consumption rates among minors. Furthermore, access to cheap alcohol in RLM is problematic as it increases the accessibility of alcohol to residents who may not be able to typically afford it.

The culture of drinking is entrenched in the lives of many community members. Furthermore, an incomplete IDP, coupled with a lack of funds and a limited focus on FASD and prenatal, gestational, and excessive alcohol consumption, hinders the improvement of the conditions within RLM. The literature on FASD-related policy development illustrates that although there has been an increase in the number of policy documents mentioning FASD in the last 10 years [31], a holistic and comprehensive policy approach is still required to address FASD.

### Limitations

This study has limitations, and the findings cannot be generalised. Firstly, the study interviewed a limited number of people for the qualitative aspect of the study and only one focus group took place in Philipstown. Thus, the interpretation of the results is limited to a small group of people in RLM. Secondly, the IDP analysis in this study is dated. It is possible that RLM has made some progress towards improving the socioeconomic conditions of the community. However, it is difficult to identify the exact improvements in the most recent IDP and LED strategies for RLM, as these documents are not publicly available.

## 5. Conclusions

Spatially, RLM experiences substantial socioeconomic challenges, an unhealthy drinking culture, economic despair, and a municipal economic strategy that fails to address FASD. A dedicated alcohol consumption census-style study is needed to broadly capture alcohol consumption in RLM. This will allow researchers to identify the exact alcohol consumption patterns and priority areas for the IDP and public health policy. RLM should more widely publicise and actively improve its policy development process so that its IDP is inclusively formulated to directly address FASD, risky and binge drinking, and prenatal and gestational alcohol consumption. The thematic analysis of the RLM IDP indicates that FASD is neglected and that a practical FASD policy development process should be instituted with input from all relevant stakeholders. The effectiveness of interventions targeting a reduction in gestational alcohol consumption should be monitored more frequently to better understand its impact.

## Figures and Tables

**Figure 1 ijerph-20-04492-f001:**
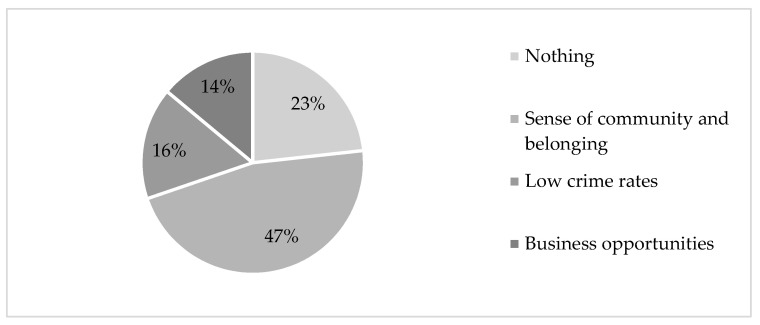
What community members appreciate about RLM.

**Figure 2 ijerph-20-04492-f002:**
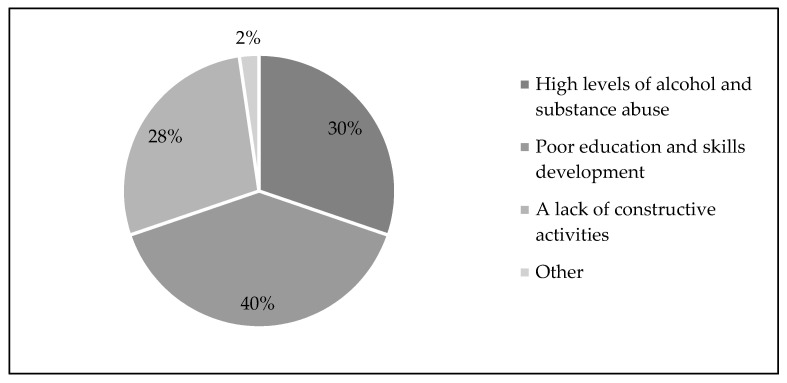
What community members do not appreciate about RLM. Source: Company A, 2017.

**Table 1 ijerph-20-04492-t001:** Summary of data sources.

	Name of Data Source	Source	Primary or Secondary Research	Publicly Available	N
1	Community Needs Assessment (1)	Statistics South Africa [24] and interviews with the communities of Petrusville and Philipstown, conducted by Company A	Secondary	No	Inconclusive
2	Community Needs Assessment (2)	Interviews with 25 persons from Petrusville and 25 persons from Philipstown, conducted by Company A	Secondary	No	50
3	In-depth Interviews	One person from Philipstown and one individual from Petrusville; interviews conducted by Company B	Secondary	No	2
4	In-depth Interviews	Three persons who implement projects in Petrusville and Philipstown; interviews conducted by authors	Primary	No	3
5	Demographic and economic data for RLM and South Africa	Statistics South Africa [24]	Secondary	Yes	7258
6	Municipal IDPs	Municipalities of South Africa [25]	Secondary	Yes	7258
7	Focus Groups	Thirty-seven residents from Petrusville and Philipstown; interviews conducted by authors	Primary	No	37

**Table 2 ijerph-20-04492-t002:** Household dynamics in RLM and SBLM.

	2011			2016		
	RLM	SBLM	SA	RLM	SBLM	SA
Number of households	2995	28,835	11,205,000	3563	35,550	16,622,000
Average household size	3.4	3.1%	3.6	3.3	3.2%	3.3
Female-headed households	34.8%	30.4%	41.2%	34.4%	32.9%	16.9%
Formal dwellings	94.7%	81.7%	77.6%	85.1%	74.8%	79.3%
Housing owned	52.3%	62.1%	53.5%	68.1%	72.4%	54.6%

Source: Municipalities of South Africa, 2022 [25,29].

**Table 3 ijerph-20-04492-t003:** Labour market in RLM and SBLM.

	2011			2021		
	RLM	SBLM	SA	RLM	SBLM	SA
Population ^1^	10,327	99,193	51,770,567	12,140	125,778	59,852,195
Working Age Population	6353	75,641	33,919,109	7258	90,463	39,302,982
Working Age Population (percentage)	61.5%	76.3%	65.5%	59.7%	71.9%	65.6%
Employed	2372	43,180	14,643,892	2597	46,868	14,541,241
Employed (percentage)	22.9%	43.5%	28.2%	21.3%	37.2%	24.2%
Unemployment Rate (official)	30.32%	14.94%	24.45%	34.49%	23.12%	33.98%
Labour Force Participation Rate	53.58%	67.11%	55.91%	53.54%	67.39%	56.49%

Source: Municipalities of South Africa 2022 [25,29]. ^1^ Working-age percentage and employed percentage were calculated as a percentage of the population.

**Table 4 ijerph-20-04492-t004:** Education in RLM and SBLM.

	2011			2016		
	RLM	SBLM	SA	RLM	SBLM	SA
No schooling	16.0%	2.4%	8.6%	11.2%	2.5%	4.9%
Secondary schooling	21.8%	28.4%	28.9%	32.2%	30.0%	28.4%
Higher education	6.6%	9.3%	11.8%	4.8%	6.3%	11.4%

Source: Municipalities of South Africa, 2022 [25,29].

**Table 5 ijerph-20-04492-t005:** Infrastructure in RLM and SBLM.

	2011			2016		
	RLM	SBLM	SA	RLM	SBLM	SA
Flushing toilet connected to a sewerage system	71.7%	92.5%	57.0%	77.4%	82.6%	60.6%
Weekly refuse removal	74.4%	96.6%	63.6%	52.2%	86.5%	61.0%
Piped water inside dwelling	53.4%	80.2%	46.3%	43.3%	74.8%	44.4%
Electricity for lighting	88.1%	97.0%	84.7%	86.3%	85.9%	90.3%

**Table 6 ijerph-20-04492-t006:** Formal employment by sector in 2018/19 in RLM.

Sector	Number of Persons in RLM	Percentage of RLM Employment ^1^
Agriculture, hunting, forestry, and fishing	795	43%
Mining and quarrying	3	0%
Manufacturing	31	2%
Electricity and gas and water supply	91	5%
Construction	52	3%
Wholesale and retail trade	164	9%
Transport; storage, and communication	35	2%
Financial, insurance, real estate, and business services	31	2%
Community, social, and personal services	335	18%
Other and not adequately defined	0	0%
Private households	226	12%
Undetermined	69	4%

Source: Municipalities of South Africa, 2022 [25]. ^1^ Calculated as the number of persons per sector as a percentage of RLM employment.

**Table 7 ijerph-20-04492-t007:** Number of institutions per sector in RLM and SBLM.

Sector	RLM	SBLM
Education	Two high schools, four primary schools, and four Early Childhood Development (ECD) centres	Twenty-eight schools, including 13 primary schools, 4 high schools, 1 combined school, 5 intermediate schools, 1 special school, and 1 school of skills.
Health	Two clinics	One district hospital, eight fixed clinics, and two satellite clinics.
Community Infrastructure	Ten sports fields, four community halls, no transport networks	Transport networks including a harbour and industrial development zone (IDZ), at least seven sports fields, and seven community halls. There are also recreational opportunities such as holiday resorts.

Source: Company A, 2017, and IDP of SBLM [30].

**Table 8 ijerph-20-04492-t008:** The results of data sources 1, 3, 4, and 7 provide insights into the drinking culture in RLM.

Themes	Community Needs Assessment (2014)	In-Depth Interviews	Focus Groups
Access to alcohol	Petrusville: “People drink cheap wine–*suurwyn* and homemade beer *soetgemmer*”.	“We buy alcohol at registered alcohol establishments, but we can also buy homemade beer for R2, this is why the vulnerable can afford it”. “Money is not an issue to drink–you can also get alcohol on credit”.	“Everyone drinks”.
Drinking culture	Petrusville: FAS, alcohol abuse (a respondent estimated that 86% of the adult population are alcoholics). Philipstown: Excess intake of alcohol begins in adolescence and has resulted in a high number of cases of FASD.	“It’s normal for us to drink when we pregnant. Hansa and Castle is good for the babies”.	“We don’t always drink with our children; children drink with their friends, and we drink with our friends”.
Income generation			“Alcohol plays an important role; every second house sells beer (as a source of income)”.

**Table 9 ijerph-20-04492-t009:** Drinking motives in RLM.

Theme	Community Needs Assessment (2014)	In-Depth Interviews	Focus Groups
Lack of recreational activities	No after-school activities for children.	“We drink because we are bored, especially over the weekend”.	“I drink because I am bored, it’s the only reason”.
Limited employment opportunities			“No opportunities, only municipal jobs”. Philipstown has no job opportunities”.
Lack of role models			“We don’t have role models, our young people must make a U-turn, and be motivated”.
Lack of infrastructure		“We have observed more infrastructure in the communities, it appears like nothing is maintained”.	
Poor human capital levels		“We have observed poor education and lack of skills”.	“Philipstown has no job opportunities”.
Link to gender-based violence or involuntary drinking	Petrusville: Most crimes are related to alcohol, for example, gender-based crimes or domestic violence. Philipstown: Increases in crime including rape are typically viewed as a consequence of alcohol misuse.		“Husbands force us to drink sometimes in Petrusville”.

**Table 10 ijerph-20-04492-t010:** Views on FASD.

	In-Depth Interviews	Focus Groups
Lack of Knowledge and Awareness	“Most mothers don’t know about the impacts of FAS (NGO A)”.	“It goes through one ear and then out the other when it comes to awareness training programs. I went through so many programs from (NGO A and government departments) at the clinics so the problem isn’t awareness, we have enough awareness”.
NGO A Presence	“I am aware of FAS, but only because I went to a course (hosted by NGO A)”. “I am aware of FAS because I stay near De Aar and my family and friends stay in De Aar. (NGO A) is present in De Aar for many years and does awareness training”.	
Perceptions of FASD	“I do think some of my family has FAS, especially when the children are rude or slow and we know that the children’s mothers drank during pregnancy”.	“We know that FAS is very high and we are well aware of high rates of FAS, nobody told us about the high rates but we can see it”. “I do know someone who has FAS–this child is 18, nobody told us before about FAS, the damage is already done. This child cannot finish school, and we force him to go to school because nobody understands”. “I cannot get any children, so I don’t care about FAS”.

**Table 11 ijerph-20-04492-t011:** IDP evaluation using Ryder’s (1996) 8-stage policy development process.

Stage and Definition	Results for RLM
Agenda Setting: Government decides that action is or is not required for a particular issue.	The agenda-setting stage of the IDP is documented in detail in its process plan chapter (pages 6–18) [14]. In addition, six community meetings were held to include feedback from the community on the development of the IDP.
Issue Filtration: Government decides on the options for action that will be formulated.	Issues in RLM were prioritised using a community and municipal priorities matrix. A sector analysis was completed and analysed. The sector analysis focuses on the spatial, social, economic, environmental, and institutional sectors [14].
Issue Definition: The exact problems and solutions are clarified.	The problems, options, and opportunities for RLM are clarified and discussed in Chapter 3 of the IDP. Areas of need and community issues are identified per sector.
Forecasting: Government considers the probable positive and negative consequences of policy implementation.	The consequences of the IDP for RLM are not discussed or highlighted in the IDP. However, in the monitoring stage, the projected impact of the projects is mentioned.
Options Analysis: Government considers all options that will best address the issue.	The IDP clearly outlines a methodology to identify and prioritise certain projects. The project design process was based on identifying a development priority, an objective, and a development strategy [14].
Objective Setting: Government sets relevant measurement indicators to track progress.	Chapter 4 of the IDP outlines the priorities, vision, and mission for RLM. In addition, projects targeting specific community problems are carefully outlined in Chapter 6 of the IDP. Each project has a logical and detailed framework for execution, which includes a budget [14].
Monitoring: Government monitors all consequences of the implemented policies (including unintended consequences).	The monitoring of project progress is mentioned throughout the IDP. However, the IDP does not reflect on progress with respect to the previous goals. However, there are sub-sections in the IDP entitled “Progress on Delivery” and “Progress on Sanitation”.
Maintenance and Succession or Termination: Government decides to continue and amend the policy or discontinue the policy entirely.	This stage is not applicable, as IDPs are mandatory for local municipalities.

## Data Availability

Demographic and economic data for RLM and South Africa and municipal IDPs are publicly available. All other data sources are not publicly available to ensure the privacy and confidentiality of the participants.

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
