# Peer review of "Rethinking Local Economic Development for Fetal Alcohol Spectrum Disorder in Renosterberg Local Municipality, South Africa"

_ijerph, 2023, doi:10.3390/ijerph20054492_

Round 1

Reviewer 1 Report

This article describes the Renosterberg Local Municipality in terms of its demographic and other mainly economic characteristics, and presents quotes from interviews and focus groups of a small sample of the population. The paper does not further scientific knowledge on this topic area, in its current format it merely reiterates that there is a problem of alcohol consumption during pregnancy in South African municipalities. In order to advance our understanding, the authors in future work may wish to explore using comparative analysis to compare RLM with another municipality that has lower rates of FASD (if this is possible), and by comparing the economic characteristics of both municipalities may be able to identify which characteristics may be associated with increased FASD. The authors also do not follow any guidelines for conducting or presenting their thematic analysis. I would recommend referring to Braun & Clark’s book Thematic Analysis: A Practical Guide.

Author Response

Dear Reviewer,

Thank you very much for your valuable input. Please find attached the requested response letter. 

Regards

Reviewer 2 Report

I appreciated the time and effort that went into the conceptualization of this paper. My suggestions below are meant to clarify and improve upon the paper, which I felt was timely, interesting, and a positive addition to the literature.

Background: 

- Please define the dop system

- In line 41, please give data on prevalence rates of FASD specific to SA, comparing it to other surveillance studies (e.g., help us understand why FASD is alarming in this area)

- I am not familiar with LED, so it would be good to have more context. Where does this occur? What outcome data is available to show this is an important approach? Basically, give context to FASD.

- Lines 57-60 give data on FASD and should be moved to the first paragraph

- Please provide more data and context to the linkages between poverty and children with FASD (line 61). There is data from the U.S. that shows that prenatal alcohol exposure is seen in higher income and highly educated people (see Patricia Price Green article, 2016). Is this not the case for SA? Provide data and appropriate citations.

- Please provide more data and context to the high economic costs of FASD (line 63). There is data about this in the U.S. (see Larry Burd's work).

Methods:

- Please clarify - this does not seem to be a mixed methods research design but instead a qualitative study with an added secondary data analysis.

- In reviewing Table 1 (and the subsequent results), it appears that this could be split up into two papers - a qualitative research paper with appropriate secondary data to triangulate the results; and a second paper focused on the secondary data collection. I see the justification for this in lines 87, but I have comments in the Results that highlight this suggestion. If you do keep this as one paper, please greatly expand the need for the secondary data that does not help confirm the qualitative data results. 

- In Section 2.2., please describe the qualitative measures more in-depth (and the instruments for the secondary data, if the authors choose to keep this as one paper)

- Please add more detail about the interviews and focus groups, including the measures, recruitment, eligibility, and consenting process. Where did the data collection occur? Who collected the data? How were they trained in qualitative data collection? 

- Section 2.3: It's unclear if this data is necessary.

- Section 2.4: Please give additional detail on qualitative data analysis. Who conducted the analysis (one person; more than one)? Was there any attempt to confirm reliability? How was the codebook developed and confirmed?

Results:

- Section 3.1: It is unclear how the data in this section relates to FASD. As noted in the Methods section, this seems like a separate paper that describes the population. It's interesting, but not clearly linked to the goal of the paper (to investigate the drinking culture and drinking motives in RLM)

- Section 3.2: This seems the focus of the paper. I appreciated the tables and how the community needs assessment was linked to the results of the qualitative data collection. 

- Table 10: Was there any data from the community needs assessment (2014) that confirms these results?

- Section 3.3: This seems like a separate paper; it did not fit clearly with the results from the qualitative data collection, which stand on their own. 

Discussion:

- Reformat based on previous suggestions. Either stay focused on the qualitative data collection and confirmation via the community needs assessment (2014) or better link the data on RLM (via the many secondary sources) to FASD and the feelings of the community. In general, the Discussion needs to be expanded greatly and should have appropriate citations added. 

Author Response

(The authors gave the same response as above.)

Reviewer 3 Report

This is a well designed paper presenting a well prepared study. Result are of high interest. 

There are two suggestions (not affecting the merit and quality of the study):

May be it is helpful to explain the dop system in a few lines. There are papers mentioning the dop such as 

Feldmann R (2020) The present and historical prevalence of fetal alcohol syndrome in children living in wine producing communities. Acta Paediatr 109: 1928-1929 doi.org/10.1111/apa.15297

or 

Adebiyi, B.O., Mukumbang, F.C. & Beytell, AM. To what extent is Fetal Alcohol Spectrum Disorder considered in policy-related documents in South Africa? A document review. Health Res Policy Sys 17, 46 (2019). https://doi.org/10.1186/s12961-019-0447-9

and others. I think Philipp May has published a lot referring to the dop here and there.

I think it may be helpful to learn something more about the "50 people" who where interviewed. You may provide a mini sample description, if possible. 

Author Response

(The authors gave the same response as above.)

Round 2

Reviewer 2 Report

I appreciate the additional work into this paper. Some additional edits to consider:

- The background is still missing some key citations on prevalence rates of FASD specific to SA, comparing it to other surveillance studies (e.g. help us understand why FASD is alarming in this area). Please add to that section specifically and add peer reviewed journal articles, instead of thesis data. 

- I again encourage the authors to include data from articles such as this one from PP Green (https://pubmed.ncbi.nlm.nih.gov/26845520/) and others that show that women at the higher end of the socioeconomic spectrum are at higher risk for alcohol consumption during pregnancy, at least in the U.S. It just gives the perspective that SES is not always correlated to drinking during pregnancy. I recommend this because of the stigma attached to drinking during pregnancy and the perpetuation that this is only a problem for those of lower SES or those of a certain race/ethnicity. 

Author Response

Response to Reviewer 2 Report 2

We appreciate the editor and reviewers for reviewing our manuscript and providing valuable comments. Due to your valued and insightful comments, we made improvements in the current version. The authors have carefully incorporated the comments into the current version of the manuscript. We hope the manuscript, after our careful revisions, meets your high standards. The authors welcome further constructive comments if any.

  1. Please add to that section specifically and add peer reviewed journal articles, instead of thesis data. May et al. (2015), Olivier (2016), Olivier (2017) added to replace thesis data on line 47.
  2. It just gives the perspective that SES is not always correlated to drinking during pregnancy. I recommend this because of the stigma attached to drinking during pregnancy and the perpetuation that this is only a problem for those of lower SES orthose of a certain race/ethnicity. Added Corrales-Gutierrez et.al 2020, Mendoza et.al 2019 (Spain), Raymond et. al (2009) UK and Anderson et. al (2014) Australia to lines 76 and 77 to show that women at the higher end of the socioeconomic spectrum can also be at higher risk of alcohol consumption during pregnancy.